# Gestational Diabetes Mellitus Treatment Schemes Modify Maternal Plasma Cholesterol Levels Dependent to Women´s Weight: Possible Impact on Feto-Placental Vascular Function

**DOI:** 10.3390/nu12020506

**Published:** 2020-02-17

**Authors:** Susana Contreras-Duarte, Lorena Carvajal, María Jesús Garchitorena, Mario Subiabre, Bárbara Fuenzalida, Claudette Cantin, Marcelo Farías, Andrea Leiva

**Affiliations:** 1Department of Obstetrics, School of Medicine, Pontificia Universidad Católica de Chile, Santiago 8330024, Chile; lpcarvaja@uc.cl (L.C.); mjgarchitorena@uc.cl (M.J.G.); mesubiabre@uc.cl (M.S.); bmfuenzalida@uc.cl (B.F.); clcantin@uc.cl (C.C.); mfarias@med.puc.cl (M.F.); 2School of Medical Technology, Health Sciences Faculty, Universidad San Sebastian, Santiago 8330024, Chile

**Keywords:** GDM, cholesterol, glycemia, GDM treatment, vascular function

## Abstract

Gestational diabetes mellitus (GDM) associates with fetal endothelial dysfunction (ED), which occurs independently of adequate glycemic control. Scarce information exists about the impact of different GDM therapeutic schemes on maternal dyslipidemia and obesity and their contribution to the development of fetal-ED. The aim of this study was to evaluate the effect of GDM-treatments on lipid levels in nonobese (N) and obese (O) pregnant women and the effect of maternal cholesterol levels in GDM-associated ED in the umbilical vein (UV). O-GDM women treated with diet showed decreased total cholesterol (TC) and low-density lipoproteins (LDL) levels with respect to N-GDM ones. Moreover, O-GDM women treated with diet in addition to insulin showed higher TC and LDL levels than N-GDM women. The maximum relaxation to calcitonin gene-related peptide of the UV rings was lower in the N-GDM group compared to the N one, and increased maternal levels of TC were associated with even lower dilation in the N-GDM group. We conclude that GDM-treatments modulate the TC and LDL levels depending on maternal weight. Additionally, increased TC levels worsen the GDM-associated ED of UV rings. This study suggests that it could be relevant to consider a specific GDM-treatment according to weight in order to prevent fetal-ED, as well as to consider the possible effects of maternal lipids during pregnancy.

## 1. Introduction

Gestational diabetes mellitus (GDM) is a pathology characterized by any degree of glucose intolerance first recognized during pregnancy [1,2]. GDM prevalence varies between 1%–25% according to the studied ethnic group and the diagnosis criteria followed [1,3]. The risk of developing GDM increases as maternal weight increases; therefore, the odds ratio for GDM is higher in obese women [4,5,6,7]. In addition, this pregnancy disease, with or without obesity, has consequences for the fetus, including pathological alterations of the feto-placental vasculature [8,9,10,11]. Additionally, GDM is associated with a higher risk of developing macrosomia [1,3,12], which is associated with an increased risk of developing cardiovascular disease (CVD) during childhood and adulthood. Furthermore, the maternal complications of GDM include an increased risk of developing endothelial dysfunction [13,14], supraphysiologic dyslipidemia [15,16], long-term type 2 diabetes mellitus (T2DM) postpartum [12,17,18] and CVD [19,20], among others. Therefore, an optimal management strategy of GDM is suggested to prevent or diminish the associated risks for both the mother and the newborn. Currently, the first-line treatment of GDM used for the optimal control of glycemia is dietary carbohydrate restriction [1,18]. If dietary treatment does not properly control glycemia, metformin and/or insulin therapy are added to the GDM management strategy [1,21]. Nevertheless, it has been described that, even after glycemic control is achieved by diet or insulin, feto-placental endothelial dysfunction is still observed in women with GDM [11,22]. This result suggests that there may be additional mechanisms other than glucose levels involved in the regulation of placental endothelial dysfunction [22]. 

Remarkably, feto-placental endothelial dysfunction has also been detected in pregnancies with supraphysiological increases in cholesterol levels [23,24,25], a condition highly prevalent in pregnant women with GDM. Interestingly, the placental vasculature lacks innervation [26] and therefore the classical vasodilators (acetylcholine and sodium nitroprusside) used in aortic or mesenteric vessels are not efficient in the placental bed. Even when calcitonin gene-related peptide (CGRP) is an unusual vasodilator in classical experiments in aortic or mesenteric vasculature, in the human placental vessels, including the umbilical vessels (veins and arteries), it is used as an endothelium-dependent relaxing agent [23,27].

Currently, the treatments for GDM are mainly focused on maintaining physiological glucose levels and an optimal maternal weight [1,28]. However, the control of other factors, such as dyslipidemia, is not considered like in patients with T2DM [29,30], and therefore is not considered as part of the GDM management strategy. Furthermore, information concerning the impact of GDM treatments on the lipid profile of mothers with GDM is indirect and controversial [31,32,33,34,35,36,37,38,39]. Therefore, the aim of this study is to determine the effect of GDM treatments on the maternal lipid profile and the impact of these changes in the GDM-associated feto-placental vascular function.

## 2. Materials and Methods

### 2.1. Study Subjects

A group of 4731 pregnant women from Hospital Clínico UC-Christus (Chile) were recruited for this retrospective and cross-sectional study. Among them, 365 women developed GDM. Lipid profiles were determined in a sample of 117 patients with GDM and 41 control (C) pregnant women.

The C and GDM women who had their lipid profiles determined during pregnancy were separated into groups, according to which pregnancy trimester samples were obtained and to body mass index (BMI). The groups were categorized as C (*n* = 41), nonobese GDM (N-GDM, *n* = 69) or obese GDM (O-GDM, *n* = 48), and the women were separated into first trimester (T1, from 0 to 14 weeks of gestation), second trimester (T2, from 14 to 28 weeks of gestation) and third trimester (T3, from 28 to 40 weeks of gestation) groups. Women with a pregestational BMI (kg/m^2^) < 30 were considered nonobese, while those with a BMI ≥ 30 were considered obese [28,40]. Umbilical cord and blood samples were collected from the women at term. Maternal age, height and fasting glycemia values were obtained in T1. In T2 insulin levels, the glycosylated hemoglobin (HbA1c) and homeostasis model assessment for insulin resistance (HOMA-IR) were determined. Oral glucose tolerance tests (OGTT) were also performed. Weight, body mass index and blood pressure data were obtained in all trimesters. Neonatal sex, gestational age, height, weight and ponderal index were determined and measured at birth. Authorized medical personnel from the patient clinical record system at the Hospital Clínico UC-Christus obtained all of the data. The reference values were determined according to the Institute of Medicine (IOM) (weight) [28], The American College of Obstetricians and Gynecologists (ACOG) (blood pressure) [41], American diabetes association (ADA) (basal glycemia, OGTT, HbA1c) [1,42,43], and the studies by Schnell et al. (insulin) [44] and Buccini and Wallace et al. (HOMA) [45,46,47,48].

For the analysis of the studied group, women with hyperthyroidism, fetal malformation, preeclampsia, hypertensive chronic syndrome, intrauterine growth restriction, insulin resistance, diabetes mellitus type 2 and/or abnormal umbilical artery Doppler results, multiple pregnancies, maternal tobacco use, alcohol or drug consumption, intrauterine infection or other medical obstetrical complications were excluded. Additionally, women who did not give informed consent or were under 18 years old were also excluded.

Our research was performed according to the Declaration of Helsinki and obtained approval from the Ethics Committee of the School of Medicine at Pontificia Universidad Católica de Chile (170803008), together with approval from the informed consent obtained from each participant. 

### 2.2. GDM Diagnosis and Treatment

GDM was diagnosed with the NICE criteria [49] between the 24th and 28th week of gestation with one of two values over the following cut-off points: fasting glycemia ≥ 100 mg/dL (5.6 mmol/L) or ≥ 140 mg/dL (7.8 mmol/L) at two hours after a 75 g glucose load [49,50].

The first treatment strategy for GDM was a reduced carbohydrate diet (1500 kcal/day and a maximum of 200 g/day of carbohydrates). If diet restriction was not enough to achieve proper glycemic control by self-monitoring, metformin (0.5–1.7 g/day) and/or insulin therapy were started [21,51]. The initial doses of insulin consisted of two injections before breakfast and bedtime of neutral protamine Hagedorn human insulin and/or injections of rapid-acting insulin before meals as needed. The recommended dosages of metformin and insulin were adjusted according to the patient’s needs, from 500 mg to 2000 mg per day. Optimal metabolic control during GDM management was monitored and recorded by each patient (delivered to the clinical staff) using serial capillary measurements of glycemia before and after meals, and glycated hemoglobin was measured in the clinical laboratory. The clinical targets of glycemic control were capillary glycemia values of 60–90 mg/dL before breakfast, 60–105 mg/dL before other meals, <140 mg/dL one hour after meals, <120 mg/dL two hours after meals and 60–99 mg/dL during the night, and HbA1c < 6.0% [50]. Plasma glucose levels were measured at T3 to confirm the achievement of glycemic control by the different treatment options.

### 2.3. Determination of Lipid Profiles

Maternal blood concentrations of total cholesterol (TC), low density lipoprotein (LDL), high density lipoprotein (HDL), very low density lipoprotein (VLDL) and triglycerides (TG) were determined in maternal brachial venous blood after 8 hours of fasting during T1 and T2 and no fasting during T3. TC, HDL and TG were determined via standard enzymatic-colorimetric assays (Cobas Integra Cholesterol (CHOLL), Cobas Integra HDL cholesterol (HDL- C), and Cobas Integra Triglycerides (TRICL) kits, Roche Diagnostic Corporation, Indianapolis, IN, USA) in a Cobas 8000 modular analyzer series (Roche Diagnostic Corporation) at the Clinical Laboratory of the Hospital Clínico UC-Christus. LDL and VLDL cholesterol were calculated from TC; HDL and TG concentrations, by applying Friedewald’s equation, as described elsewhere [23].

### 2.4. High Cholesterol Cut-Off Value

For umbilical vein reactivity assays, normal total cholesterol (NTC) or high total cholesterol (HTC) levels were defined as being over a cut-off point of 280 mg/dL at the term of pregnancy, based on the literature [23,24,27,52,53,54,55,56].

### 2.5. Umbilical Vein Reactivity

Ring segments of 2–4 mm in length were dissected from human umbilical veins in phosphate buffered solution (PBS) (130 mmol/L NaCl, 2.7 mmol/L KCl, 0.8 mmol/L Na2HPO4, 1.4 mmol/L KH2PO4, pH 7.4, 4 °C). Vein rings were mounted in a myograph (610M Multiwire Myograph System, Danish Myo Technology A/S, Denmark) for isometric force measurements in a Krebs physiological solution (118.5 mmol/L NaCl, 4.7 mmol/L KCl, 25 mmol/L NaHCO3, 1.2 mmol/L MgSO4, 1.2 mmol/L KH2PO4, 2.5 mmol/L CaCl2, 5.5 mmol/L D-glucose, 37 °C, pH 7.4) bubbled with a mixture of 95% O_2_/5% CO_2_. The optimal diameter for each vessel was determined by the maximal active response to 65 mmol/L KCl [25]. UV optimal diameters for N and GDM women were comparable between groups (4.82 ± 0.25 and 5.54 ± 0.41, respectively). All the experiments were performed in vessels at the optimal diameter.

Endothelium-dependent relaxation was evaluated as the response to calcitonin gene-related peptide (CGRP, 0.001-1000 nmol/L, 5 minutes) (Sigma-Aldrich) in KCl- preconstricted vessels (32.5 mmol/L because at this concentration the vasodilatory response of the umbilical vein is detectable and subjected to modulation) in the absence or presence of 100 µmol/L NG-nitro-L-arginine methyl ester (L-NAME) and a nitric oxide synthase (NOS) inhibitor (20 minutes). Changes in isometric tension were recorded using the software LabChart (LabChart 7 for Windows, ADInstruments, Australia) coupled to a PowerLab data acquisition system (PowerLab 8/30 Data Acquisition System, ADInstruments, Australia). Maximal relaxation (R_max_) of the vessels was determined as the percentage of relaxation caused by CGRP compared to the maximal contraction. The data were graphed as the difference between the response to CGRP in the absence of L-NAME and the response to CGRP in the presence of the NOS inhibitor (NOS-dependent relative response). However Appendix A shows maximal relaxation with or without L-NAME in controls and GDM groups. The concentration of the drug required the production of 50% of the maximum relaxation (EC_50_) and was calculated and tabulated.

### 2.6. Data Analysis

Clinical data are presented as the mean ± S.D. For vascular reactivity assays, the values are presented as the mean ± S.E.M. The normality of the data was determined with the Kolmogorov-Smirnov test. Comparisons between two groups were performed using Student’s unpaired t-tests and Mann-Whitney tests for parametric and nonparametric data, respectively. The differences between more than two groups were determined by the analysis of variance (ANOVA) and Friedman’s tests for parametric and nonparametric data, respectively. If ANOVA or Friedman’s test demonstrated a significant interaction between variables, post hoc analyses were performed by the multiple-comparison Bonferroni’s or Dunn’s correction tests, respectively. A chi-square test was performed to analyze categorical variables. Correlations were performed using Pearson’s or Spearman’s correlation coefficient for parametric or nonparametric data, respectively. A *p* value < 0.05 was considered significant for all of the tests. Statistical tests were performed with GraphPad Prism 7.0 statistical software (GraphPad Software Inc., San Diego, CA, USA).

## 3. Results

### 3.1. Clinical Characteristics of the Participants

The GDM prevalence was 7.7% in the studied population during the evaluated period (365/4731).

Age was comparable between groups, and the women with GDM were shorter in height compared to the women in group C (Table 1).

All pregnant women increased in weight during pregnancy, and patients in the O-GDM group presented higher weights during pregnancy than women in groups C and N-GDM (Table 1). Concerning weight gain, both groups with GDM gained less weight than group C. Additionally, women in the O-GDM group gained less weight than women in the N-GDM group (Table 1).

As for weight, BMIs increased in all pregnant women during pregnancy, and women in the O-GDM group showed higher BMIs than women in groups C and N-GDM (Table 1).

Mean arterial pressure was normal (< 106 mmHg) in all groups during pregnancy (Table 1).

Fasting glucose determined in trimester 1 was below normal levels (< 100 mg/dL) in all groups (Table 1).

In the OGTTs, fasting glucose levels were normal in all the groups (< 100 mg/dL). However, glucose levels were higher in women in groups N-GDM and O-GDM than in group C (Table 1).

The two-hour glucose levels after a 75 g glucose load were above normal levels (< 140 mg/dL) in all women with GDM and higher compared to women in group C (Table 1).

Insulin and HbA1c levels were only determined in women with GDM. These values were normal (range 2.6–24.9 µUI/ml for insulin and < 6.0% for HbA1c), even though women in the O-GDM group showed higher levels of insulin than women in the N-GDM group (Table 1). Additionally, HOMA-IR was above normal levels (> 2.6) in women in the O-GDM group relative to women in the N-GDM group (Table 1).

Newborn variables such as sex, gestational age, height, birth weight and ponderal index were similar across all groups (Table 1).

### 3.2. Effects of Obesity and GDM on the Lipid Profile of Pregnant Women

TC, TG, HDL, LDL and VLDL were measured in group C and in women with GDM during pregnancy (Figure 1). As shown in Figure 1A, 1C and 1D, during T1 no changes were observed between groups C, N-GDM and O-GDM concerning TC (184.6 ± 33.5, 183.5 ± 36.1 and 184.7 ± 25.4 mg/dL, respectively), HDL (66.3 ± 12.8, 63.8 ± 13.1 and 61.9 ± 14.8 mg/dL, respectively) or LDL (104.2 ± 30.6, 94.6 ± 28.7 and 93.4 ± 15.4 mg/dL, respectively) levels, respectively. However, an increase in TG was observed in the O-GDM group compared to group C (151.1 ± 64.1 versus 102.7 ± 39.8 mg/dL) (Figure 1B); a similar effect was also observed in VLDL levels (30.2 ± 12.8 versus 20.6 ± 7.9 mg/dL) (Figure 1E). In T2, TC levels in the O-GDM group were lower than those in group C (203.5 ± 36.6 versus 237.7 ± 38.6 mg/dL) (Figure 1A). This change was associated with lower levels of HDL (66.6 ± 17.1 versus 79.1 ± 22.4 mg/dL) (Figure 1C) and LDL (99.1 ± 27.9 versus 125.2 ± 32.1 mg/dL) in the O-GDM group compared to group C (Figure 1D). Moreover, TG levels were higher in the N-GDM and O-GDM groups than in group C (205.7 ± 94.5, 189.1 ± 60.3 and 167.9 ± 59.9 mg/dL, respectively) (Figure 1B). VLDL levels were higher in the N-GDM and O-GDM groups than in group C (41.1 ± 18.9, 37.8 ± 12.0 and 33.1 ± 12.3 mg/dL, respectively) (Figure 1E). In T3, TC and HDL in the N-GDM and O-GDM groups were lower than those in group C (TC: 244.9 ± 46.1, 231.8 ± 49.1 and 263.1 ± 48.2 mg/dL, respectively; HDL: 66.2 ± 18.8, 65.6 ± 17.1 and 76.3 ± 18.8 mg/dL, respectively) (Figure 1A, 1C). LDL were lower in the O-GDM group compared to group C (122.2 ± 40.8 versus 137.9 ± 37.3 mg/dL) (Figure 1D). No significant changes were observed in TG or VLDL in the N-GDM or O-GDM groups compared with group C (TG: 229.4 ± 86.3, 224.8 ± 71.0 and 241.5 ± 84.2 mg/dL, respectively; VLDL: 45.9 ± 17.2, 44.4 ± 13.3 and 48.6 ± 23.5 mg/dL, respectively) (Figure 1B, 1E).

### 3.3. Correlation Between Maternal BMI and Lipid Levels

There was no correlation between BMI and TC, TG, HDL, LDL or VLDL levels at T1, T2 or T3 in group C or groups with GDM (Table 2).

### 3.4. Effect of GDM Treatments on Glycemic Control

The treatments for GDM were only diet (14.1%), diet plus metformin (13.3%) or insulin (53.5%), or insulin plus metformin (19.1%), depending on the criteria established by clinical staff and the achievement of metabolic targets. In total, 15.7% of the N-GDM group used diet as treatment compared to 11.6% of the O-GDM group. A higher percentage of the N-GDM group (18.4%) used diet plus metformin compared to 7% of the O-GDM group. In total, 55.3% of the N-GDM group used diet plus insulin compared to 53.5% of the O-GDM group. A lower percentage of the N-GDM group received the three treatments together (10.5%) compared to 27.9% in the O-GDM group (Figure 2A).

Glycemic control at the end of pregnancy was achieved under the different treatment options, and the results were similar between the N-GDM and O-GDM groups (Figure 2B). No significant differences were observed in non-fasting glucose levels in T3 for women in N-GDM and O-GDM groups treated with diet (91.9 ± 16.4 and 96.3 ± 14.0 mg/dL, respectively), in those who used diet and metformin (98.4 ± 39.6 and 92 ± 3.0 mg/dL, respectively), in women who used diet plus insulin (80.8 ± 8.5 and 84.7 ± 12.0 mg/dL, respectively), or in those who used diet, metformin and insulin (80.8 ± 9.1 and 85.4 ± 13.5 mg/dL, respectively) (Figure 2B).

### 3.5. Effect of GDM Treatment on Lipid Levels

Lipid levels were determined in women with GDM at T3 with groups categorized according to their GDM treatment, as discussed in Section 3.4.

Treatment with diet was associated with lower TC levels in the O-GDM group compared to the N-GDM group (220.7 ± 54.4 versus 270.5 ± 40.1 mg/dL) (Figure 3A) as well as lower LDL levels (119.3 ± 26.6 versus 158.8 ± 35.7 mg/dL) (Figure 3B). In the N-GDM group compared to the O-GDM group, no additional changes in TC (253.4 ± 37.1 versus 185.6 ± 21.5 mg/dL) or LDL levels (130.3 ± 30.5 versus 84.2 ± 29.8 mg/dL) were observed in women treated with diet plus metformin with respect to diet alone. Interestingly, treatment with diet plus insulin was associated with augmented TC levels in the O-GDM group with respect to the N-GDM group (250.4 ± 50.5 versus 236.0 ± 45.7 mg/dL) as well as increased LDL levels (134.5 ± 46.1 versus 128.7 ± 40.2 mg/dL) (Figure 3B). Furthermore, in women treated with diet plus metformin and insulin, TC and LDL levels were comparable in both the N-GDM and O-GDM groups (TC: 234.3 ± 65.2 versus 206.1 ± 32.3 mg/dL; LDL: 116.3 ± 21.7 versus 105.4 ± 26.9 mg/dL) (Figure 3A, 3B).

No changes were observed between groups with GDM and their received treatments in HDL (Figure 3C), VLDL (Figure 3D) or TG (Figure 3E).

Additional analysis of the same data showed that the different treatments had differentiated effects on the lipid profile in the N-GDM or O-GDM groups. Thus, in the N-GDM group, treatment with diet plus insulin was associated with lower TC levels compared to those treated with diet (236.0 ± 45.7 versus 270.5 ± 40.1 mg/dL) (Figure 3A). This result was associated with decreased LDL levels in those in the N-GDM group treated with diet plus insulin and treated with diet plus insulin and metformin with respect to those treated with diet (128.7 ± 40.2, 116.3 ± 21.7 and 158.1 ± 35.7 mg/dL, respectively) (Figure 3B). In regards to TC in the O-GDM group, higher levels were determined in those in the O-GDM group treated with diet plus insulin compared to those treated with diet (250.4 ± 50.5 versus 220.7 ± 54.4 mg/dL) without changes in LDL or other lipoproteins.

### 3.6. Effect of GDM and Maternal Lipids on Human Umbilical Vein Reactivity

Vascular reactivity was evaluated in human umbilical vein rings from women in groups C and N-GDM with NTC or HTC at term (Table 3). All groups were comparable in maternal age, height, weight, weight gain, BMI, mean arterial pressure and glycemia during the first trimester (Table 3). OGTT results were different in the N-GDM (NTC and HTC) group compared to group C (NTC and HTC). TC was increased in both HTC groups compared to NTC groups (Table 3).

The vascular rings were challenged to relaxation with CGRP in a dose-dependent manner. Maximal relaxation (R_max_) in the N-GDM (including NTC and HTC) group was lower than that in the C (including NTC and HTC) group (18.3 ± 1.9 versus 32.1 ± 5.2%) (Figure 4A, Table 4). Moreover, the half maximal effective concentration (EC_50_) in response to CGRP in the vein rings from the N-GDM group was comparable to that of group C (7.4 ± 0.16 versus 7.4 ± 0.18 nM) (Figure 4A, Table 4).

Interestingly, the R_max_ of NOS-dependent relaxation in the UV rings from the women in group C with NTC was higher than those determined to have HTC in group C (39.1 ± 6.7 versus 16.7 ± 2.0%) (Figure 4B, Table 4). In addition, the R_max_ in women from the N-GDM group with NTC was higher than those determined to have HTC in the N-GDM group (28.7 ± 2.4 versus 15.3 ± 1.8%) (Figure 4B, Table 4). No differences were observed between women in the group C with NTC and women in the N-GDM group with NTC (39.1 ± 6.7 versus 28.7 ± 2.4) (Figure 4B, Table 4). However, the R_max_ in women from group C with HTC was higher than that of women from the N-GDM group with HTC (16.7 ± 2.0 versus 15.3 ± 1.8%) (Figure 4B, Table 4). Accordingly, the EC_50_, in response to CGRP in vein rings from the women in group C with HTC, was higher than that of the women in group C with NTC (8.4 ± 0.2 versus 7.6 ± 0.2nM) (Figure 4B, Table 4). The EC_50_ from women in the N-GDM group with HTC was lower than that determined in women from the N-GDM and C groups with NTC (6.5 ± 0.2, 10.7 ± 0.3 and 7.6 ± 0.2 nM, respectively) (Figure 4B, Table 4). Finally, the EC_50_ from women in the N-GDM group with HTC was lower than that of the women from group C with HTC (6.5 ± 0.2 versus 8.4 ± 0.2 nM).

## 4. Discussion

The current study proposes, for the first time, the relevance of making a differentiation in the GDM treatment according to the maternal weight state, in order to recommend a specific treatment for each group. Additionally, this investigation reinforces the importance of measuring lipids during pregnancy as a routine exam. However, this study has some limitations related to the small sample size in each subgroup and the cross-sectional nature of the study, which limited the interpretation of the results.

The prevalence of GDM was 7.7% in our study group. This investigation was conducted in a sample of adequately managed women with GDM in terms of recommended gestational weight gain [28] and glycemic control [1], based on three therapeutic options, including only diet or diet plus metformin and/or insulin [21,36,57,58]. The lipid profile of women in the N-GDM and O-GDM groups changed during pregnancy. However, women developed dyslipidemia from T1 in the O-GDM group and from T2 in the N-GDM group, which in both cases persisted until T3, and these changes were independent of BMI. While all GDM subgroups of treatment options showed proper metabolic control when evaluated at T3, we observed differential modulation of TC and LDL levels between obese and nonobese patients with GDM. Moreover, in those of the N-GDM group with normal glucose and high TC levels, vasodilation of the umbilical vein rings was impaired compared to that of women with normal TC levels. These data suggest that GDM treatments differentially modulate TC levels in nonobese and obese pregnant women, which could be an important factor in modulating feto-placental endothelial function.

Concerning the prevalence of GDM, the result of this study was comparable to those described in other groups from America (8% in Chile and United States of America) and other continents (2–18% in Asia, 0–14% in Africa and 15% in Europe) [21,59,60,61,62,63,64].

The HOMA-IR values were higher in the O-GDM group compared to the N-GDM group. While there are no cut-off values for HOMA-IR during pregnancy and only one report suggests that GDM is associated with HOMA-IR values over 2.6 during T1 [44], our findings suggest that, in GDM, obesity could be associated with increased insulin resistance, which is a concept widely recognized in nonpregnant populations [65,66,67,68].

It is well documented that newborns from women with GDM present macrosomia as a consequence of hyperinsulinemia [69,70]. However, macrosomia was absent in the offspring of this studied group. This phenomenon could be related to the normal glucose, insulin and TG levels observed at term in all women of the studied groups due to the assigned GDM treatments received during pregnancy [21,70,71,72].

Regarding the lipid profile, all of the studied groups showed increased TC, LDL and TG levels during pregnancy, as previously reported for the women in group C and women with GDM [15,16]. This physiological change is required to fulfill all of the needs of the developing fetus [25,30,73,74,75,76,77,78,79,80,81,82].

Nevertheless, a complex dyslipidemia state seems to be observed during pregnancy in the O-GDM group. Our results confirm the previous description of higher levels of TG and therefore of VLDL during T1 in the O-GDM group and during T2 in the N-GDM group [83,84,85]. The abnormal rise of TG in women with GDM is produced by high serum estrogen concentrations and augmented insulin resistance that generate hypertriglyceridemia [86]. Here, this result was evident in women from the O-GDM group from T1 and T2 who presented high HOMA-IR. While the literature broadly described women with GDM as being associated with increased TG levels during T3 [83,87,88,89,90,91,92], in this study, a normalization of this lipid level was observed, likely due to GDM treatments in both groups of women with GDM, which is in agreement with other reports [84].

Reports of low levels of HDL at T2 and T3 in women with GDM, especially in O-GDM, have been previously reported [89,90,92,93,94]. The reduction in HDL levels could likely be due to an exchange of TG for cholesterol ester in the core lipids of HDL [84]. We suggest that a decreased level of HDL is an important issue to consider since the cardio-protective functions of HDL could be impaired in these women [95].

Regarding LDL and in line with our results, some studies described reduced TC and LDL levels in obese women with GDM compared to C women [84,93,96]. Other studies showed increased levels of TC in obese women with GDM with respect to C women due to an increase in LDL [83,91]. In our O-GDM group, the reduced levels of TC and LDL were not related to maternal BMI (Table 2); thus, we propose that the changes are related to the chosen treatment for GDM (diet, metformin and/or insulin). Interestingly, inclusion of metformin for insulin-treated women with GDM was associated with reductions in TC and LDL during the last trimester of pregnancy, an effect that was not observed in insulin-treated patients.

Concerning GDM treatments, in this study, the most frequently used treatment was insulin in the N-GDM and O-GDM groups, in contrast to treatments used in other centers worldwide where the main treatment is diet [43,97]. Women in the O-GDM group received higher proportions of the three interventions to achieve proper glycemic levels, in comparison to women in the N-GDM group. The effect of different GDM treatments on the lipid profile was evaluated. It was found that TG, VLDL and HDL levels were comparable between the N-GDM and O-GDM groups under the different treatments. However, TC and LDL levels were differentially modulated by the different treatments. In the O-GDM group relative to the N-GDM group, TC and LDL were decreased in those treated with diet, increased in those treated with insulin and comparable in those who received all three treatments. In this regard, scarce information is known about the impact of GDM treatments on lipid regulation because these treatments are rarely measured as routine controls during pregnancy [16,30]. The decreased TC and LDL levels of women in the O-GDM group treated with diet could be due to changes in lipid metabolism, as described in obese patients with T2DM treated with diet [98]. The increase in plasma LDL levels produced by insulin therapy in women of the O-GDM group (Figure 3) has been previously reported in women with GDM [12,85]. However, the mechanisms associated with this change are largely unknown [85,99]. Since insulin resistance is higher in our O-GDM group compared to the N-GDM group, and since it has been shown that insulin resistance associates with increased cholesterol synthesis favoring the production of VLDL, and therefore its metabolization to LDL via the cholesteryl ester transfer protein (an enzyme whose activity increases during pregnancy) [100,101,102], we propose that cholesterol metabolism could be further altered in women of the O-GDM group who were treated with insulin. This phenomenon was not observed in women treated with insulin-sensitizing metformin (Figure 3), which could also decrease glycation levels present in LDL, as it occurs in patients with T2DM [103], thereby allowing normal binding between LDLs and their receptors and then decreasing LDL levels in plasma. While it is suggested that insulin therapy produces an improvement in lipid levels in women with GDM [37,38,93], other studies have also suggested that insulin therapy generates a pro-atherogenic profile in overweight and obese women with GDM, which is associated with increased levels of TC and TG, low levels of HDL and apo AI (the main protein of HDL), increased levels of the inflammatory marker interleukin-6 [12,14,85,99], impaired blood flow in the forearm skin [14,104] and greater arterial stiffness after delivery [105]. Therefore, in some women, insulin therapy could worsen the maternal cardiovascular profile [99].

It is widely recognized that GDM is associated with impaired endothelial-dependent relaxation of umbilical vein rings and feto-placental endothelial dysfunction [106]. Interestingly, these alterations are still present in the feto-placental vasculature from women with GDM and are treated with diet or insulin [11,22], suggesting that factors other than glucose may be involved in the pathophysiology of GDM. We propose that maternal TC levels could be one of the possible factors contributing to this phenomenon. In line with this hypothesis, our results showed reduced endothelial-dependent relaxation of umbilical vein rings from pregnant women with GDM with increased levels of TC (levels comparable to those described in the O-GDM group treated with diet plus insulin, Figure 3), in comparison to rings from women with GDM with normal TC levels. These results show, for first the time, that maternal TC levels have an impact on the feto-placental endothelial dysfunction associated with GDM, although the participating mechanisms remain unknown. We suggest that increased TC levels could worsen feto-placental endothelial function by modifying nitric oxide availability, which has been previously shown in rings from the placental vessels from pregnant women without GDM, but increased TC levels [23,24,25,27]. Therefore, we propose that increased levels of TC, as described in obese women with GDM treated with insulin, could impair feto-placental endothelial function.

In addition to the possible involvement of the nitric oxide signaling in this study, it would be interesting to evaluate the nitric oxide metabolites and the oxidative stress in these samples as well, since these pathways are affected in GDM and dyslipidemic pregnancies [54,107,108,109]. However, what is happening with these parameters when they are present at the same time is still unknown.

Finally, and regarding the use of CGRP in our experiments of vascular reactivity, even though CGRP is an unusual vasodilator in classical experiments in aortic or mesenteric vessels, in the human umbilical vessels (veins and arteries), which is a vascular bed that lacks innervation [26], it is used as an endothelium-dependent relaxant agent [110]. 

CGRP is an aminoacid neuropeptide [111] that acts through a seven transmembrane domain G protein-coupled receptor, calcitonin receptor-like receptor (CRLR), which has three ligands: adrenomedullin, intermedin and CGRP. In addition, three receptor activity modifying proteins: RAMP1, RAMP2, and RAMP3. Coexpression of CRLR with RAMP1 forms a CGRP receptor [112]. In the human placenta, both CRLR and RAMP1 are expressed in the endothelium and underlying smooth muscle cells in the umbilical, chorionic and stem villous vessels [110], suggesting that CGRP may play a role in the control of feto-placental vascular tone [113], and therefore is a physiologic vasodilator in these vascular beds. Actually, it has been shown that chronic administration of CGRP antagonist CGRP8-37 to pregnant rats caused a significant reduction in pup weight and increased systolic blood pressure and fetal mortality rate [114], and these effects were dose dependent, suggesting that CGRP may be involved in the control of feto-placental circulation [113].

Similar experiments were performed with Insulin as a vasodilator, but GDM endothelial response was impaired as we expected likely do to a dysregulation in the glucose metabolism (Appendix A). Noticeable, when NTC- and HTC-C vascular reactivity was performed in placenta as well, but using Sodium nitroprusside (SNP) as a nitric oxide (NO) donor, no changes were observed between control groups with normal and high cholesterol levels, suggesting that the mechanisms independent of the NO machinery are intact (Appendix A).

## 5. Conclusions

In summary, our data suggest that GDM treatments differentially modulate TC levels in nonobese and obese pregnant women, which could be an important factor modulating feto-placental endothelial function. Therefore, we propose that TC modulation should be considered an additional target for GDM management strategies. Thus, GDM therapy selection should be performed according to the maternal nutritional state and lipid profile to avoid further potential impairment of the feto-placental endothelial function (Figure 5).

GDM treatments are focused in control glycemia and weight but not lipids, which are also dysregulated in these patients. In this study, all the schemes used to treat the disease succeed in the control of glycemia. However, lipid levels were not well controlled and this control was weight-dependent and lipid specific, i.e., TC and LDL were improved in O-GDM respect to N-GDM when these women were treated with Diet and the opposite effect was observed when these patients were treated with Diet + Insulin. Furthermore, it has been described that when glucose levels are controlled, but TC is elevated in N and N-GDM women, the feto-placental vasculature is impaired. Nevertheless, when glucose and TC are corrected, feto-placental vasculature could be improved. Abbreviations: GDM: Gestational Diabetes Mellitus; N: Nonobese; O: Obese; LDL: Low Density Lipoprotein; TC: Total Cholesterol; HTC: High Total Cholesterol; NTC: Normal Total Cholesterol Dashed lines correspond to our proposal.

## Figures and Tables

**Figure 1 nutrients-12-00506-f001:**
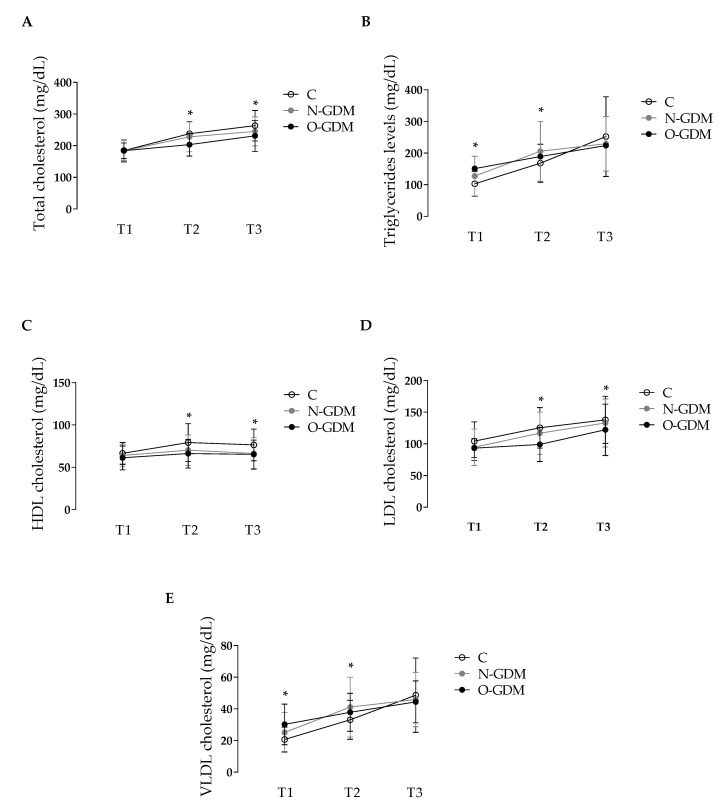
Lipid profile during pregnancy in the control group and nonobese or obese women with gestational diabetes mellitus. Lipid determination was performed in the control women (C, *n* = 41) group and nonobese (N, *n* = 69) or obese (O, *n* = 48) women with gestational diabetes mellitus (GDM). Lipid determination was performed at the 1st (T1, 0–14 weeks of gestation), 2nd (T2, 14–28 weeks of gestation) and 3rd trimester (T3, 28–40 weeks of gestation) of pregnancy. (**A**) TC: total cholesterol, (**B**) TG: triglycerides, (**C**) HDL: high density lipoproteins, (**D**) LDL: low density lipoproteins, (**E**) VLDL: very low density lipoproteins, T: trimester. Values are the mean ± S.D. and units are mg/dL. Significant differences were considered with **p* < 0.05.

**Figure 2 nutrients-12-00506-f002:**
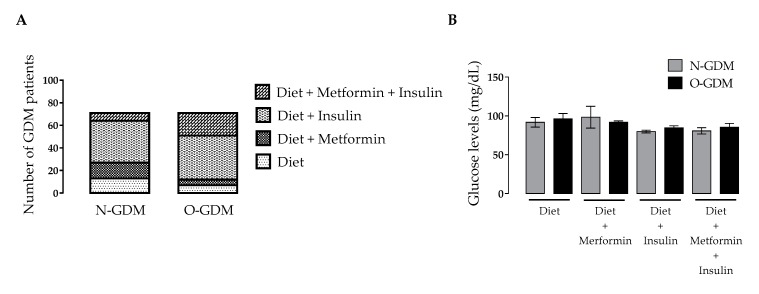
Effect of gestational diabetes mellitus treatment on glucose levels in nonobese and obese pregnant women at term. Glucose levels were determined in nonobese (N, *n* = 69) or obese (O, *n* = 48) women with gestational diabetes mellitus (GDM) pregnancy. (**A**) Frequency of treatments in women with GDM, χ^2^ = 4.263, *p* = 0.039 for diet plus metformin and χ^2^ = 6.259, *p* = 0.012 for diet plus insulin and metformin. (**B**) Glucose levels determined during the 3rd trimester after different GDM treatments. GDM treatments were as follows: diet, diet plus metformin, diet plus insulin and diet plus insulin and metformin. Values are the mean ± S.D. Significant differences were considered with **p* < 0.05.

**Figure 3 nutrients-12-00506-f003:**
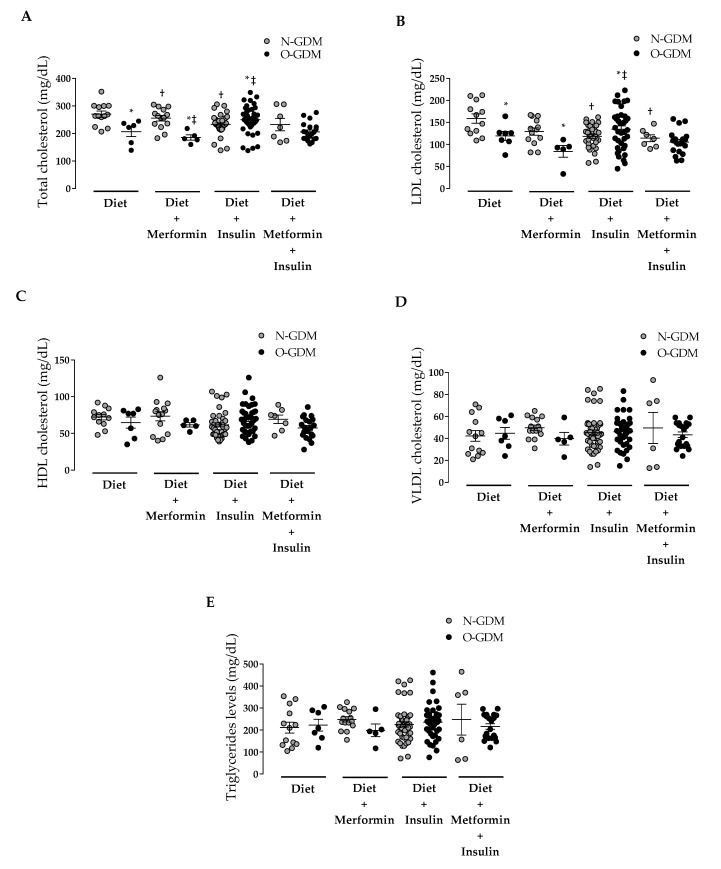
Effect of different treatments for gestational diabetes mellitus on maternal lipid levels in nonobese and obese pregnant women at term. Lipid determination was performed at the 3rd trimester in nonobese (N, *n* = 50) or obese (O, *n* = 46) women with gestational diabetes mellitus (GDM). GDM treatments were diet, diet plus metformin, diet plus insulin and diet plus insulin and metformin. (**A**) Total cholesterol levels, (**B**) low density lipoproteins cholesterol levels (LDL), (**C**) high density lipoproteins cholesterol levels (HDL), (**D**) triglycerides levels and (**E**) very low density lipoproteins cholesterol levels (VLDL). Values are the mean ± S.D. Significant differences were considered with *p* < 0.05. * respect to the N-GDM group, † respect to diet in the N-GDM group, ‡ respect to diet in the O-GDM group.

**Figure 4 nutrients-12-00506-f004:**
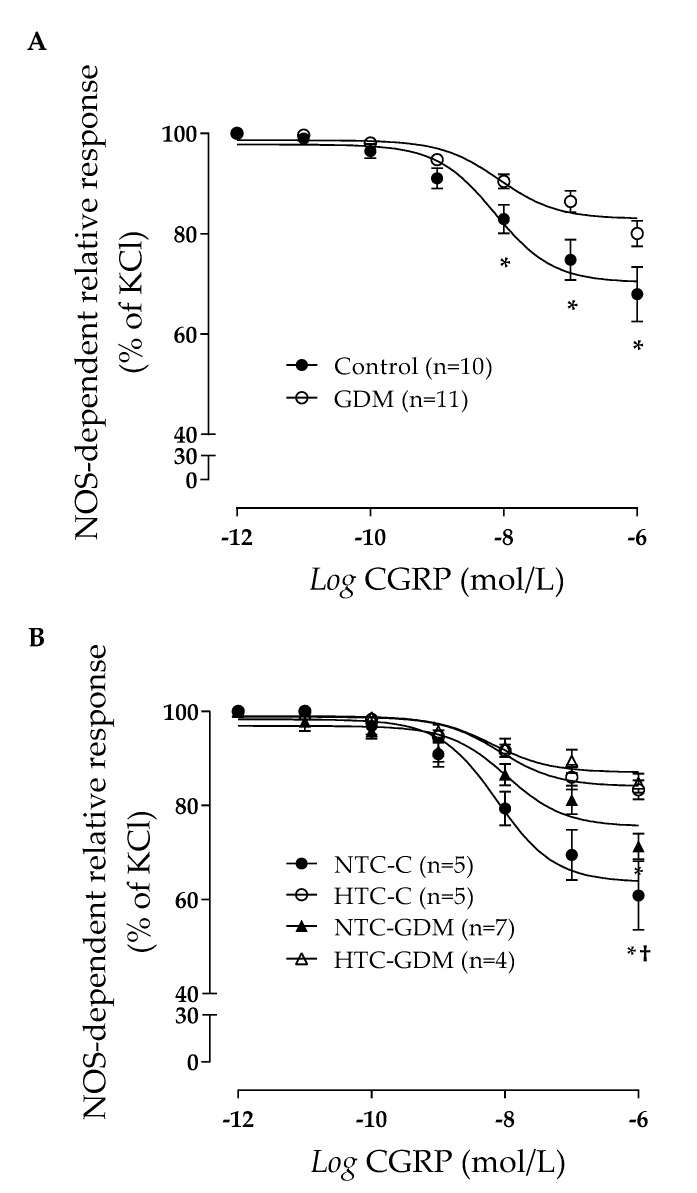
Impact of high total cholesterol levels on the dilation of human umbilical vein rings from nonobese control women and women with gestational diabetes mellitus. Human umbilical vein rings were obtained at term from nonobese pregnant women from the control (C) group or those with gestational diabetes mellitus (GDM). The women were categorized as having normal total cholesterol (NTC, TC < 280 mg/dL) or high total cholesterol (HTC, TC ≥ 280 mg/dL) levels. (**A**) Relaxation was determined in umbilical vein rings from the C (*n* = 10) or GDM (*n* = 11) groups in response to calcitonin gene-related peptide (CGRP, 0.001–1000 nmol/L, 5 minutes). Rings were pre-constricted with 32.5 mmol/L KCl in the absence or presence of 100 µmol/L NG-nitro-L-arginine methyl ester (L-NAME, 20 minutes), and the percentage of inhibition of maximal dilation was graphed. (**B**) The same samples from A were categorized as NTC-C (*n* = 5), HTC-C (*n* = 5), NTC-GDM (*n* = 7) or HTC-GDM (*n* = 4) and vascular dilation was assayed like that in A. Values are mean ± S.E.M. Significant differences were considered with *p* < 0.05. * respect to C or NTC-C, † respect to HTC-C.

**Figure 5 nutrients-12-00506-f005:**
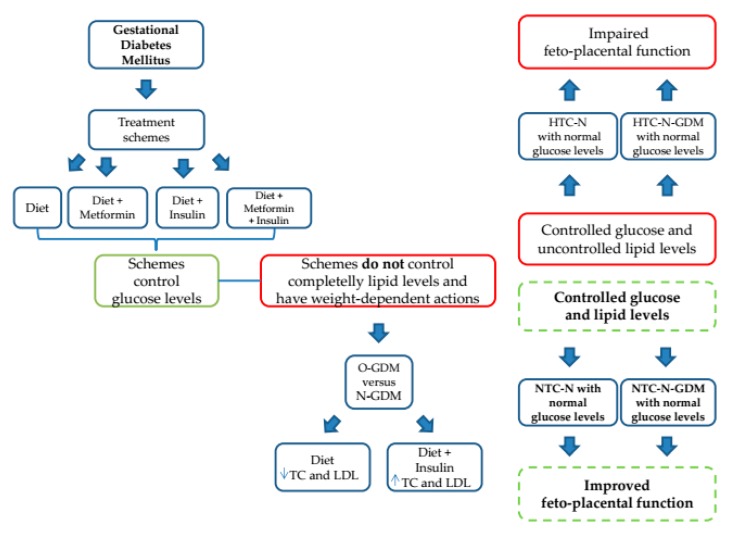
Schematic summary.

**Table 1 nutrients-12-00506-t001:** Clinical characteristics of pregnant women and newborns.

Variables			C (*n* = 41)	N-GDM (*n* = 69)	O-GDM (*n* = 48)
**Maternal variables**					
	Age (years)		31.7 ± 3.8 (23–36)	34.8 ± 4.8 (25–55)	34.6 ± 4.5 (26–42)
	Height (cm)		164 ± 0.1 (152–175)	160 ± 0.1 * (149–173)	160 ± 0.1 *^,^† (148–177)
	Weight (Kg)				
		T1	62.8 ± 5.8 (53–74)	60.9 ± 7.0 (43.3–77)	81.3 ± 7.8 *^,^† (68.8–96)
		T2	68.8 ± 5.2 (59–78)	64.9 ± 7.1 * (45–82.8)	82.0 ± 9.6 *^,^† (64–115)
		T3	75.2 ± 4.4 ‡ (68–83)	68.8 ± 7.2 *^,^‡ (51.9–88.5)	85.2 ± 10.0 *^,^†^,^‡ (66–120)
	Weight gain (Kg)		12.4 ± 4.8	8.0 ± 3.4 *	5.8 ± 3.8 *^,^†
			(4–22)	(0.2–12)	(0.2–13.5)
	BMI (Kg/m^2^)				
		T1	23.5 ± 1.8 (20.5–1.8)	24.0 ± 2.3 (17.3–28.9)	31.5 ± 2.3 *^,^† (30–38.9)
		T2	25.7 ± 1.5 (22–29.3)	25.6 ± 2.2 (18–30)	32.0 ± 3.1 *^,^† (28.9–37.3)
		T3	28.1 ± 2.0 ‡ (25.7–29.9)	26.9 ± 2.0 *^,^‡ (20.8–29.8)	33.4 ± 3.4 *^,^†^,^‡ (30.1–41.4)
	Mean arterial pressure (mm Hg)				
		T1	81.5 ± 5.9 (73.3–93.3)	81.7 ± 7.1 (63.3–95.6)	79.1 ± 5.6 † (67.3–86.7)
		T2	82.2 ± 5.9 (73.3–93.3)	88.4 ± 9.7 * (66.7–103.3)	80.5 ± 7.5 † (66.7–93.3)
		T3	86.4 ± 5.3 ‡ (73.3–96.7)	83.1 ± 6.3 * (66.7–101.7)	81.4 ± 7.5 * (70–95)
	Fasting glycemia (mg/dL)		84.0 ± 4.0 (80–88)	85.5 ± 10.9 (63–103)	84.3 ± 8.4 (69–98)
	OGTT (mg/dL)				
		Basal glycemia	77.5 ± 4.3 (69–85)	81.8 ± 8.4 * (70–110)	87.2 ± 14.2 *^,^† (70–129)
		Glycemia at 2 hours	106.4 ± 17.4 (75–139)	157.7 ± 15.7 * (140–201)	154.7 ± 17.3 * (141–200)
	Insulin (µUI/ml)		-	9.5 ± 5.1 (2.7–27.4)	11.9 ± 6.5 * (3.3–26.6)
	HbA1c %		-	5.7 ± 2.0 (2.4–14.4)	5.9 ± 1.4 (4.9–13.5)
	HOMA-IR		-	2.0 ± 1.5 (0.5–5.2)	3.1 ± 2.8 * (0.8–12.9)
**Newborn variables**					
	Sex (female/male)		25/16	37/32	21/27
	Gestational age (weeks)		38.5 ± 1.0 (37–40)	38.6 ± 3 (36–40)	38.2 ± 3.2 (37–40)
	Height (cm)		49.5 ± 1.7 (45–52.5)	49.9 ± 2.0 (46–52)	49.5 ± 1.5 (48–52)
	Birth weight (grams)		3308 ± 382 (2600–4060)	3286 ± 412 (2430–3830)	3243 ± 405 (2700–3950)
	Ponderal index (grams/cm^3^ x100)		2.6 ± 0.2 (2.35–3.02)	2.7 ± 0.2 (2.43–3.14)	2.7 ± 0.3 (2.3–3.16)

Control (C, *n* = 41) and nonobese (N) or obese (O) women with gestational diabetes mellitus (GDM, *n* = 117) were included. Clinical variables at the 1st trimester (T1, 0–14 weeks of gestation), 2nd trimester (T2, 14–28 weeks of gestation) or 3rd trimester (T3, 28–40 weeks of gestation) of pregnancy were tabulated. Weight, body mass index (BMI) and mean arterial pressure were determined for all groups at T1, T2 and T3. Maternal age, height and fasting glycemia were determined in all groups at T1. The insulin, glycosylated hemoglobin (HbA1c) and homeostatic model assessment for insulin resistance (HOMA-IR) levels were determined, and the oral glucose tolerance test (OGTT) was performed in all GDM groups at T2. At birth, neonatal sex, gestational age, height, weight and ponderal index were determined. Normal values or ranges for the measured variables are as follows: mean arterial pressure < 106 mmHg, OGTT basal glycemia < 100 mg/dL, OGTT glycemia at 2 hours <140 mg/dL, insulin 2.6-24.9 µUI/ml, HbA1c < 6.0% and HOMA-IR until 2.6. Data are presented as the mean ± S.D. (range). Significant differences were considered with *p* <0.05. * respect to C, † respect to the N-GDM group, ‡ respect to T1.

**Table 2 nutrients-12-00506-t002:** Correlation between lipid levels and body mass index during pregnancy in women with gestational diabetes mellitus.

**T1**																				
Lipid	TC		TG					HDL				LDL					VLDL			
Group	C	GDM	C			GDM		C		GDM		C		GDM			C		GDM	
N° pairs	23	27	23			27		23		27		23		27			23		27	
Spearman r	0.08	0.19	0.37			0.32		-0.04		-0.32		0.32		0.21			0.37		0.37	
*P*	0.703	0.342	0.085			0.101		0.843		0.102		0.137		0.292			0.082		0.07	
**T2**																				
Lipid	TC			TG					HDL				LDL					VLDL		
Group	C	GDM		C	GDM				C		GDM		C		GDM			C	GDM	
N° pairs	40	31		40	31				37		31		37		31			36	31	
Spearman r	−0.01	−0.04		0.05	0.30				0.20		−0.02		−0.09		−0.07			−0.22	0.30	
*P*	0.951	0.826		0.742	0.096				0.237		0.934		0.590		0.699			0.196	0.097	
**T3**																				
Lipid	TC			TG					HDL				LDL					VLDL		
Group	C	GDM		C			GDM		C		GDM		C			GDM		C		GDM
N° pairs	41	44		41			44		41		44		41			44		41		44
Spearman r	0.30	−0.17		−0.07			0.22		0.12		−0.09		−0.19			−0.18		−0.13		0.17
*P*	0.061	0.271		0.655			0.151		0.451		0.577		0.221			0.254		0.434		0.276

Control (C) women or women with gestational diabetes mellitus (GDM) were included. Spearman correlation analyses between body mass index (BMI) and lipid levels during the 1st trimester (T1, 0–14 weeks of gestation), 2nd trimester (T2, 14–28 weeks of gestation) or 3rd trimester (T3, 28–40 weeks of gestation) of pregnancy were performed. For each trimester, the following is shown: Group, pairs correlated, values for Spearman r and *P.* TC: total cholesterol, TG: triglycerides, HDL: high density lipoproteins, LDL: low density lipoproteins, VLDL: very low density lipoproteins. Significant differences were considered with *p* < 0.05.

**Table 3 nutrients-12-00506-t003:** Clinical characteristics of pregnant womem and newborns recruited for feto-placental vascular reactivity assays.

Variables				NTC-C (*n* = 5)	HTC-C (*n* = 5)	NTC-GDM (*n* = 7)	HTC-GDM (*n* = 4)
**Maternal variables**							
	Weeks of gestation			39.8 ± 0.4 (39–40)	38.4 ± 1.1 * (37–40)	38.6 ± 0.8 * (38–40)	39.3 ± 0.5 (39–40)
	Age (years)			31.4 ± 8.0 (19–38)	28.8 ± 3.5 (27–34)	30.9 ± 5.6 (23–39)	34.3 ± 3.6 (29–37)
	Height (cm)			1.65 ± 0.1 (1.59–1.73)	1.61 ± 0.1 (1.51–1.67)	1.60 ± 0.04 (1.55–1.64)	1.63 ± 0.1 (1.57–1.78)
	Weight (Kg)						
		T1		63.2 ± 6.4 (52–68)	59.9 ± 7.1 (52.1–71)	61.7 ± 5.9 (53–70)	67.8 ± 2.1 (66–70)
		T3		73.2 ± 6.9 ‡ (63–80)	69.9 ± 5.0 ‡ (63–74)	71.0 ± 5.9 ‡ (65–78)	74.2 ± 1.0 ‡ (73.5–76)
	Weight gain (Kg)			10.0 ± 5.1 (3–16)	9.0 ± 6.0 (1.5–10.5)	7.6 ± 2.9 (5–11)	7.8 ± 1.7 (6–10)

	BMI (Kg/m^2^)						
		T1		23.3 ± 2.2 (21–26)	23.0 ± 2.0 (21–25)	24.4 ± 2.2 (21–27)	25.7 ± 3.3 (21–28)
		T3		26.1 ± 0.9 ‡ (25–27)	27.1 ± 0.9 ‡ (26–28)	27.7 ± 1.5 ‡ (25–29)	28.3 ± 2.9 (24–29.8)
	Mean arterial pressure (mm Hg)						
		T1		76.1 ± 5.4 (70–83)	82.2 ± 5.1 (77–87)	79.8 ± 6.6 (73–90)	76.7 ± 4.7 (73–83)
		T3		89.3 ± 11.5 (73–105)	82.2 ± 5.9 (73–90)	85.9 ± 9.5 (73–103)	80.2 ± 3.7 (77–84)
	Fasting glycemia (mg/dL)			81.8 ± 4.0 (77–86)	77.0 ± 9.8 (70–88)	86.6 ± 5.7 (80–91)	80.3 ± 3.6 (75–83)
	OGTT (mg/dL)						
		Basal glycemia		73.0 ± 7.0 (65–78)	70.5 ± 0.7 (70–71)	77.1 ± 5.0 (70–85)	79.0 ± 5.7 (73–86)
		Glycemia at 2 hours		104.3 ± 10.0 (93–112)	98.0 ± 35.4 (73–123)	148.7 ± 4.7 * (144–155)	164.3 ± 20.8 * (146–192)
		Insulin (µUI/ml)		-	-	13.6 ± 3.0 (11.5–17)	11.5 ± 7.3 (2.5–17)
		HbA1c %		-	-	4.9 ± 0.3 (4.6–5.3)	5.4 ± 1.7 (4.5–9.9)
		HOMA-IR		-	-	2.5 ± 0.4 (2.2–2.9)	2.3 ± 1.4 (0.7–4.2)
	Lipids (mg/dL)						
		Total cholesterol	223 ± 29.2 (187–258)		308 ± 22.3 * (292–347)	232 ± 19.8 (202–256)	337 ± 44.5*^,^† (298–376)
		Triglycerides	276 ± 45.4 (201–317)		235 ± 96.3 (184–394)	219 ± 73.3 (153–331)	268 ± 76.6 (172–-335)
		High density lipoproteins	61 ± 14.5 (49–84)		53 ± 8.0 (47–67)	72 ± 14.0 (50–91)	83 ± 23.3 (50–110)
		Low density lipoproteins	119 ± 18.1 (101–142)		208 ± 25.2 * (166–234)	147 ± 27.5 (109–176)	159 ± 48.6 (113–220)
		Very low density lipoproteins	43 ± 24.4 (13–63)		47 ± 18.7 (31–79)	46 ± 13.2 (33–66)	54 ± 15.5 (34–67)
**Newborn variables**							
	Sex (female/male)			2/3	1/4	4/3	2/2
	Height (cm)			51.6 ± 1.2 (50–53)	50.2 ± 1.3 (49–53)	50.3 ± 1.7 (47–52)	50.6 ± 1.4 (49–52)
	Birth weight (grams)			3500 ± 199.1 (3320–3810)	3220 ± 346.5 (2890–3800)	3287 ± 312.1 (2640–3610)	3545 ± 438.5 (2900–3860)
	Ponderal index (grams/cm^3^ × 100)			2.5 ± 0.1 (2.4–2.7)	2.6 ± 0.1 (2.4–2.7)	2.6 ± 0.2 (2.3–2.8)	2.7 ± 0.3 (2.5–3.1)

Control (C) women with normal total cholesterol (NTC) (NTC-C, *n* = 5), high total cholesterol (HTC) (HTC-C, *n* = 5), women with gestational diabetes mellitus (GDM) and NTC (NTS-GDM, *n* = 7) and with GDM and HTC (HTC-GDM, *n* = 4) were included. Clinical variables at the 1st trimester (T1, 0–14 weeks of gestation), 2nd trimester (T2, 14–28 weeks of gestation) or 3rd trimester (T3, 28–40 weeks of gestation) of pregnancy were tabulated. Weight, body mass index (BMI) and mean arterial pressure were determined in all groups at T1, T2 and T3. Maternal age, height and fasting glycemia were determined in all groups at T1. Insulin, glycosylated hemoglobin (HbA1c), oral glucose tolerance test (OGTT) and homeostatic model assessment for insulin resistance (HOMA-IR) were performed in all GDM groups at T2. Maternal lipids were determined at T3. At birth, neonatal sex, gestational age, height, weight, and ponderal index were determined. Normal values or ranges for the measured variables are as follows: mean arterial pressure < 106 mmHg, OGTT basal glycemia ≤ 100 mg/dL, OGTT glycemia at 2 hours ≤ 140 mg/dL, insulin 2.6–24.9 µUI/ml, HbA1c < 6.0%, HOMA-IR until 2.6, and HTC ≥ 280 mg/dL. Data are presented as the mean ± S.D. (range). Significant differences were considered with *p* < 0.05. * respect to NTC-C, ‡ respect to T1, † respect to NTC-GDM.

**Table 4 nutrients-12-00506-t004:** The impact of maternal high cholesterol levels on dilation of human umbilical vein rings.

Parameter	CGRP					
Group	Control (*n* = 10)	GDM (*n* = 11)	NTC-C (*n* = 5)	HTC-C (*n* = 5)	NTC-GDM (*n* = 7)	HTC-GDM (*n* = 4)
**EC_50_ average (nM)**	7.4 ± 0.18	7.4 ± 0.16	7.6 ± 0.2	8.4 ± 0.2 *	10.7 ± 0.3 *^,^†	6.5 ± 0.2 *^,^†^,^‡
**R_max_ average (%)**	32.1 ± 5.2	18.3 ± 1.9 *	39.1 ± 6.7	18.2 ± 2 *	28.7 ± 2.4 †	15.3 ± 1.8 *^,^†

Nonobese Control (C, *n* = 10) women or those with gestational diabetes mellitus (GDM, *n* = 11) were included. The women were categorized as having normal total cholesterol (NTC, TC < 280 mg/dL) or high total cholesterol (HTC, TC ≥ 280 mg/dL) levels. Human umbilical vein rings were obtained at term from the different groups. CGRP, calcitonin gene-related peptide. EC_50_, half maximal effective concentration. R_max_, maximal relaxation. Values are the mean ± S.E.M. (*n* = 4–7). Significant differences were considered with *p* < 0.05. * respect to the C or NTC-C groups, † respect to the HTC-C group, ‡ respect to the NTC-GDM group.

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
