# Peer review of "Gestational Diabetes Mellitus Treatment Schemes Modify Maternal Plasma Cholesterol Levels Dependent to Women´s Weight: Possible Impact on Feto-Placental Vascular Function"

_nutrients, 2020, doi:10.3390/nu12020506_

Round 1
Reviewer 1 Report
The study was thoroughly developed. The methods and statistical analysis of the results were described in detail. However, the graphic presentation of the results is inappropriate. Figures and axis description are too small. They should be corrected, especially Figs. 1-3. The legend for Tab. 3 is combined with a description of the results. What does the † symbol mean? (total cholesterol in Table 3). The form of the results presentation should be changed. The conclusions are based on the results.
Author Response
We are thankful to this reviewer for her/his comments, we took all the suggestions and we improved the presentation of the results as it was asked.
“Figures and axis description are too small. They should be corrected, especially Figs. 1-3”.
Thank you for the comment, the changes were made in these figures as it was suggested.
“The legend for Tab. 3 is combined with a description of the results”.
We improved the legend and now is similar to legend of Table 1; please see the page 14, lines 332 to 348 of the manuscript.
“What does the † symbol mean? (total cholesterol in Table 3)”.
Thank you for the observation, the symbol † correspond to the significant differences respect to NTC-GDM group. The description was incorporated in the legend of this table. Please check line 348 of the manuscript to see the correction.
“The form of the results presentation should be changed”
We apologize if the way in which we have presented the results does not seem appropriate and we hope you can suggest us what would be the best way to present them please. So far, to improve the presentation of the results, we have improved the figures presentation as it was suggested before.
Reviewer 2 Report
This manuscript describes the relationship lipid and endothelial function in gestational diabetes mellitus. The authors investigated the effect of GDM treatments on the maternal lipid profile and CGRP-induced relaxation response of umbilical vein. The authors found that maternal TC levels have an impact on the feto-placetal endothelial dysfunction associated with GDM. Although the data were interesting, the authors should address some concerns as following.
Firstly, the manuscript, as a whole, the language and abbreviation need to be checked again.
The title should be revised. Revise from “GDM” to “Gestational diabetes mellitus”.
Abstract line 20. Revise exactly state of the C women.
Introduction: Please shortly describe an importance of CGRP-induced relaxation as endothelium-dependent relaxation in UV?
In Methods 2.5, please describe more details. Please state diameter of UV. Absolute force of KCl-induced contraction. Why did the authors used 32.5 mmol/L of KCL? The authors used an inhibitor of NOS, and y-axis show about NOS-dependent relative response, however, how determine the value? Please describe in details (Figure 4).
In Figure 4, the authors should present representative traces in the absence and presence of NOS inhibitor. Please show the data of CGRP-induced relaxation in the absence and presence of NOS inhibitor. What were other dilators (e.g., ACh, or nitric oxide donor)-induced relaxations, and contractile responses? The authors should discuss an importance of CGRP in UV. What were nitric oxide metabolites and oxidative stress markers levels in the plasma?
Please make a schematic summary based on the present data.
Author Response
We are thankful to this reviewer for her/his comments and for highlighting the fact that the data reported in the manuscript is interesting. We have analysed the comments/suggestions, followed all and raised specific replies for each of them.
“The manuscript, as a whole, the language and abbreviation need to be checked again”.
Language was checked again, this time by a native colleague and his corrections were included in the text. Please check changes along the manuscript.
In addition, we include the certificate obtained by AJE English language editorial with verification code: A5E6-0930-EE76-F204-8426.
Abbreviations were reviewed as well and the corrections were included.
Please check line 22 of the manuscript to see the correction.
“The title should be revised. Revise from “GDM” to “Gestational diabetes mellitus”.
The title was reviewed and the suggestion was incorporated, then the final title of the manuscript is: Gestational Diabetes Mellitus treatment schemes modify maternal plasma cholesterol levels dependent to women´s weight: possible impact on feto-placental vascular function.
Abstract line 20. Revise exactly state of the C women.
Thank you for the observation, effectively it was a mistake in the abbreviation, we replace it by: N, from nonobese.
Please check line 22 of the manuscript to see the correction.
Introduction: Please shortly describe an importance of CGRP-induced relaxation as endothelium-dependent relaxation in UV?
The description we added to the introduction of the manuscript is the following:
Interestingly, the placental vasculature lacks innervation (Marzioni D et al, 2004) and therefore the classical vasodilators (acetylcholine, sodium nitroprusside) used in aortic or mesenteric vessels are not efficient in the placental bed. Even when calcitonin gene-related peptide (CGRP) is an unusual vasodilator in classical experiments in aortic or mesenteric vasculature, in the human placental vessels including the umbilical vessels (veins and arteries), it is used as an endothelium-dependent relaxant agent (Leiva A et al 2013; Fuenzalida B et al, 2018).
Please check lines 52 to 57 of the manuscript to see the description.
- Marzioni, D.; Tamagnone, L.; Capparuccia, L.; Marchini, C.; Amici, A.; Todros, T.; Bischof, P.; Neidhart, S.; Grenningloh, G.; Castellucci, M. Restricted innervation of uterus and placenta during pregnancy: evidence for a role of the repelling signal Semaphorin 3A. Dev Dyn 2004, 231, 839-848.
- Leiva, A.; de Medina, C.; Salsoso, R.; Sáez, T.; San Martín, S.; Abarzúa, F.; Farías, M.; Guzmán-Gutiérrez, E.; Pardo, F.; Sobrevia, L. Maternal hypercholesterolemia in pregnancy associates with umbilical vein endothelial dysfunction: role of endothelial nitric oxide synthase and arginase II. Arterioscler Thromb Vasc Biol 2013, 33, 2444-2453.
- Fuenzalida, B.; Sobrevia, B.; Cantin, C.; Carvajal, L.; Salsoso, R.; Gutiérrez, J.; Contreras-Duarte, S.; Sobrevia, L.; Leiva, A. Maternal supraphysiological hypercholesterolemia associates with endothelial dysfunction of the placental microvasculature. Sci Rep 2018, 8, 7690.
These references are included in the respective section of the introduction and references.
“In Methods 2.5, please describe more details. Please state diameter of UV”.
The maximum contraction point is defined as the ideal baseline stretch or optimal diameter (Lew MJ & McPherson GA, 1996). The stretch needed to reach the optimal diameter depends on the caliber of each vessel. Additionally, it is described that basal physiological tension of blood vessels in the organism is closer to the optimal length (Burkholder TJ & Lieber RL, 2001; Winters TM et al, 2011). Therefore, this parameter allows inferring differences in the diameter of each vessel under physiological conditions.
The changes in the optical diameter were included in the new version of the manuscript in this section according to the suggestions made previously by the reviewer. However, no changes were observed between groups.
“UV optimal diameters for N and GDM women were comparable between groups (4.82 ± 0.25 and 5.54 ± 0.41, respectively). All the experiments were performed in vessels at optimal diameter.”
Please check lines 141-143 of the manuscript to see the correction.
- Lew, MJ.; McPherson, GA.. Isolated tissue techniques. In: The Pharmacology of Vascular Smooth Muscle. Garland CJ, Angus J (eds) Oxford University Press, Oxford 1996, 25-41
- Burkholder, TJ.; Lieber, RL. Sarcomere length operating range of vertebrate muscles during movement. J Exp Biol 2001, 204, 1529-1536.
- Winters, TM.; Takahashi, M.; Lieber, RL.; Ward, SR. Whole muscle length-tension relationships are accurately modeled as scaled sarcomeres in rabbit hindlimb muscles. J Biomech 2011, 44, 109-115.
“Absolute force of KCl-induced contraction. Why did the authors used 32.5 mmol/L of KCL?”
This is an important comment; we used 32.5 mM KCl because at this concentration the vasodilatory response of the umbilical vein is detectable and subjected to modulation. In line to our results obtained using this KCl concentration, there are other reports in human umbilical vein rings where we have used similar KCl concentration to our present study (Guzmán-Gutiérrez et al, 2012; Krause et al, 2012 Leiva et al, 2015; Fuenzalida B et al, 2018). However, there are several studies available regarding dilation of human umbilical vein rings that were preconstricted with higher concentrations of KCl (e.g., 60 mM in Caliskan et al, 2006; 70 mM in García-Huidobro et al, 2007; 62.5 mM in Westermeier et al, 2011) compared with what we have used in the present study (i.e., 32.5 mM KCl). In fact the use of 60 mM KCl as a preconstrictor has been shown not to alter the vasodilation of human umbilical vein rings to propofol (a potent sedative), a phenomenon that most likely depends of Ca2+-activated K+ channels (K+Ca2+). Furthermore, other studies in human chorionic veins have used higher KCl concentrations as well (75 mM, Cruz et al, 1998). Thus, it is suggested that under preconstriction caused by higher KCl concentration compared to what we have used here, the vasodilatory response of this type of human vessels it is also detectable and subjected to modulation.
- Guzmán-Gutiérrez, E.; Westermeier, F.; Salomón, C.; González, M.; Pardo, F.; Leiva, A.; Sobrevia L. Insulin-increased L-arginine transport requires A (2A) adenosine receptors activation in human umbilical vein endothelium. PLoS One 2012, 7, e41705.
- Krause, BJ.; Prieto, CP.; Muñoz-Urrutia, E.; San Martín, S.; Sobrevia, L.; Casanello P. Role of arginase-2 and eNOS in the differential vascular reactivity and hypoxia-induced endothelial response in umbilical arteries and veins. Placenta 2012, 33, 360-366.
- Leiva, A.; Fuenzalida, B.; Westermeier, F.; Toledo, F.; Salomón, C.; Gutiérrez, J.; Sanhueza, C.; Pardo, F.; Sobrevia, L. Role for tetrahydrobiopterin in the fetoplacental endothelial dysfunction in maternal supraphysiological hypercholesterolemia. Oxid Med Cell Longev 2015, 2015, 5346327.
- Fuenzalida, B.; Sobrevia, B.; Cantin, C.; Carvajal, L.; Salsoso, R.; Gutiérrez, J.; Contreras-Duarte, S.; Sobrevia, L.; Leiva, A. Maternal supraphysiological hypercholesterolemia associates with endothelial dysfunction of the placental microvasculature. Sci Rep 2018, 8, 7690.
- Caliskan, E.; Kayhan, Z.; Tufan, H. Propofol inhibits potassium chloride induced contractions of isolated human umbilical vessels. Eur J Anaesthesiol 2006, 23, 411-417.
- García-Huidobro, DN.; García-Huidobro, MT.; Huidobro-Toro, JP. Vasomotion in human umbilical and placental veins: role of gap junctions and intracellular calcium reservoirs in their synchronous propagation. Placenta 2007, 28, 328-338.
- Westermeier, F.; Salomón, C.; González, M.; Puebla, C.; Guzmán-Gutiérrez, E.; Cifuentes, F.;Leiva, A.; Casanello, P.; Sobrevia, L. Insulin restores gestational diabetes mellitus-reduced adenosine transport involving differential expression of insulin receptor isoforms in human umbilical vein endothelium. Diabetes 2011, 60, 1677-1687.
- Cruz, MA.; Gallardo, V.; Miguel, P.; Carrasco, G.; González, C. Mediation by 5-HT2 receptors of 5-hydroxytryptamine-induced contractions of human placental vein. Gen Pharmacol 1998, 30, 483-488.
“The authors used an inhibitor of NOS, and y-axis show about NOS-dependent relative response, however, how determine the value? Please describe in details (Figure 4).”
We used L-NAME (Nitro-L-arginine methyl ester hydrochloride) as eNOS inhibitor as it is described in methods section of the manuscript. What we did with the obtained data, were substract the values from the experiments with L-NAME (eNOS-dependent response) to those without L-NAME and this is what we have plotted in Figure 4.
In Figure 4, the authors should present representative traces in the absence and presence of NOS inhibitor. Please show the data of CGRP-induced relaxation in the absence and presence of NOS inhibitor.
Here we show the figure corresponding to grouped Control and GDM vascular reactivity data treated with L-NAME ( same groups showed in Figure 4A on the manuscript).
(Figure, not possible to copy here, we so sorry)
What were other dilators (e.g., ACh, or nitric oxide donor)-induced relaxations, and contractile responses?
In this specific manuscript we used insulin as well, but the endothelial response was impaired as we expected because GDM women present a dysregulation in the glucose metabolism, this is the reason why we did not include these results to the manuscript. Here we show the result we obtained.
(Figure, not possible to copy here, we so sorry)
Concerning to the use of acetylcholine, since our investigation uses the placenta as model of investigation that lacks innervation; firstly, acetylcholine, that acts in NO-dependent and independent way (Doyle, MP. & Duling, BR., 1997), avoiding the endothelium response, cannot be used to resolve our question that it was mainly related to the effect of the lipids on the endothelium because of their signaling mechanisms. Noticeable, when NTC- and HTC-C vascular reactivity was performed in placenta as well, but using SNP as a NO donor, no changes between groups were observed, suggesting that the mechanisms independent of the NO machinery are intact (see figure below).
(Figure, not possible to copy here, we so sorry)
- Doyle, MP.; Duling, BR. Acetylcholine induces conducted vasodilation by nitric oxide-dependent and -independent mechanisms. Am J Physiol 1997, 272, H1364- H1371.
The authors should discuss an importance of CGRP in UV.
The paragraph we added to the discussion section of the manuscript is the following:
Finally and regarding the use of CGRP in our experiments of vascular reactivity, even though CGRP is an unusual vasodilator in classical experiments in aortic or mesenteric vessels, in the human umbilical vessels (veins and arteries), a vascular bed that lacks innervation (26), it is used as an endothelium-dependent relaxant agent (112).
CGRP is an aminoacid neuropeptide (113) that acts through a seven transmembrane domain G protein-coupled receptor, calcitonin receptor-like receptor (CRLR), which has three ligands: adrenomedullin, intermedin and CGRP. In addition, three receptor activity modifying proteins: RAMP1, RAMP2, and RAMP3. Coexpression of CRLR with RAMP1 forms a CGRP receptor (114). In the human placenta, both CRLR and RAMP1 are expressed in the endothelium and underlying smooth muscle cells in the umbilical, chorionic, and stem villous vessels (112), suggesting that CGRP may play a role in the control of feto-placental vascular tone (115), and therefore is a physiologic vasodilator in these vascular beds. Actually, it has been shown that chronic administration of CGRP antagonist CGRP8-37 to pregnant rats caused a significant reduction in pup weight and increased systolic blood pressure and fetal mortality rate (116), and these effects were dose dependent, suggesting that CGRP may be involved in the control of feto-placental circulation (115).
26. Marzioni, D.; Tamagnone, L.; Capparuccia, L.; Marchini, C.; Amici, A.; Todros, T.; Bischof, P.; Neidhart, S.; Grenningloh, G.; Castellucci, M. Restricted innervation of uterus and placenta during pregnancy: evidence for a role of the repelling signal Semaphorin 3A. Dev Dyn 2004, 231, 839-848.
Dong, YL.; Vegiraju, S.; Gangula, PR.; Yallampalli, C. Involvement of CGRP in control of human fetoplacental vascular tone. Am J Physiol Heart Circ Physiol 2004, 286, H230 –H239. Rosenfeld, MG.; Mermod, JJ.; Amara, SG.; Swanson, LW.; Swchenko, PE.; Rivier, J.; Vale, WW.; Evans, RM. Production of a novel neuropeptide encoded by the calcitonin gene via tissue-specific RNA processing. Nature 1983, 304, 129 –135. Roh, J.; Chang, CL.; Bhalla, A.; Klein, C.; Hsu, SYT. Intermedin is a calcitonin/ calcitonin gene-related peptide family peptide acting through the calcitonin receptor-like receptor/receptor activity-modifying protein receptor complexes. J Biol Chem 2004, 279, 7264 –7274. Dong, YL.; Green, KE.; Vegiragu, S.; Hankins, GD.; Martin, E.; Chauhan, M.; Thota, C.; Yallampalli, C. Evidence for Decreased Calcitonin Gene-Related Peptide (CGRP) Receptors and Compromised Responsiveness to CGRP of Fetoplacental Vessels in Preeclamptic Pregnancies. J Clin Endocrinol Metab 2005, 90, 2336-2343. Gangula, PRR.; Dong, YL.; Wimalawansa, SJ.; Yallampalli, C. Infusion of pregnant rats with calcitonin gene-related peptide (CGRP)(8 –37), a CGRP receptor antagonist, increases blood pressure and fetal mortality and decreases fetal growth. Biol Reprod 2002, 67, 624- 629.
Please check lines 502-517 in the discussion section of the manuscript. In addition, all the references cited here are included in the manuscript in the corresponding section.
What were nitric oxide metabolites and oxidative stress markers levels in the plasma?
This is a very relevant question for this study and it is the continuation of this research line. Unfortunately, we did not measure those parameters in these patients. However, in a current investigation that we are performing (the continuation of this study), we measured the L-citrulline levels (which is the metabolite released with nitric oxide in the same proportion when the endothelial nitric oxide synthase (eNOS) is active and in healthy conditions) by HPLC technique in the same four groups that we used in Table 3. Similar to what we found in vascular reactivity results that we showed in this manuscript, L-citrulline levels were decreased in HTC-C (1.3±0.7 AU, P<0.007) and HTC-GDM (0.5±0.3 AU, P<0.05) and increased in NTC-GDM (3.6±1.7 AU, P<0.002) relative to NTC-C (1±0.4 AU). This result is in line with the vascular reactivity experiments, suggesting and validating the impairment we suggested before in the feto-placental vasculature when lipids are not corrected in the mother. This result is going to be published in another manuscript that we are currently writing.
Regarding to the oxidative stress markers in the plasma, it would have been very interesting to have these results, but we did not measure them, and this information is no published in the literature as far as we know. But what we would have expected to find, as it is described broadly in the literature, is that women with GDM have presented altered/high oxidative stress (lipid peroxides) in plasma (Chen X & Scholl TO, 2005; Lappas M et al, 2011) and also in the feto-placental vasculature (Sun Y, 2018) and in their newborns (Grissa O et al, 2007) respect to normal groups. However, when these groups are categorized by their lipids levels, those GDM women, with normal lipid levels, should have lower oxidative stress markes, compared to GDM women with high lipid levels, following the same line in normal pregnancies with altered lipids, as it was described by Liguori A et al, 2007.
In the discussion section, we included a small paragraph, highlighting the relevance of this idea:
“In addition to the possible involvement of the nitric oxide signaling in this study, it would be interesting to evaluate the nitric oxide metabolites and the oxidative stress in these samples as well, since these pathways are affected in GDM and dyslipidemic pregnancies (Liguori A et al 2007; Chen X & Scholl TO, 2005; Lappas M et al, 2011; Sun Y, 2018). However, what is happening with these parameters when they are present at the same time, is still unknown.”
Please check lines 498 to 501 to see this comment.
- Lappas, M.; Hiden, U.; Desoye, G.; Froehlich, J.; Hauguel-de Mouzon, S.; Jawerbaum, A. The role of oxidative stress in the pathophysiology of gestational diabetes mellitus. Antioxid Redox Signal 2011, 15, 3061-3100.
- Chen, X.; Scholl, TO. Oxidative stress: changes in pregnancy and with gestational diabetes mellitus. Curr Diab Rep 2005, 5, 282-288.
- Sun, Y.; Kopp, S.; Strutz, J.; Gali, CC.; Zandl-Lang, M.; Fanaee-Danesh, E.; Kirsch, A.; Cvitic, S.; Frank, S.; Saffery, R.; Björkhem, I.; Desoye, G.; Wadsack, C.; Panzenboeck, U. Gestational diabetes mellitus modulates cholesterol homeostasis in human fetoplacental endothelium. Biochim Biophys Acta Mol Cell Biol Lipids 2018, 1863, 968-979.
- Grissa, O.; Atègbo, JM.; Yessoufou, A.; Tabka, Z.; Miled, A.; Jerbi, M.; Dramane, KL.; Moutairou, K.; Prost, J.; Hichami, A.; Khan, NA. Antioxidant status and circulating lipids are altered in human gestational diabetes and macrosomia. Transl Res 2007, 150, 164-171.
- Liguori, A.; D'Armiento, FP.; Palagiano, A.; Balestrieri, ML.; Williams-Ignarro, S.; de Nigris, F.; Lerman, LO.; D'Amora, M.; Rienzo, M.; Fiorito, C.; Ignarro, LJ.; Palinski, W.; Napoli, C. Effect of gestational hypercholesterolaemia on omental vasoreactivity, placental enzyme activity and transplacental passage of normal and oxidised fatty acids. BJOG 2007, 114, 1547-1556.
The first three references were included in the manuscript.
Please make a schematic summary based on the present data.
Thank you for this important request, the following scheme summarizes the investigation we have performed. In the manuscript you can also find the legend of this scheme (Figure 5 in the conclusion section).
(scheme, not possible to copy here, we so sorry) Please see the file.
